# Sustainability Assessment Based on Integrating EKC with Decoupling: Empirical Evidence from China

Donghui Lv [1], Ruru Wang [2] and Yu Zhang [2,*]

1    School of Finance, Jilin University of Finance and Economics, Changchun 130117, China; lvdh@jlu.edu.cn
2    School of Geographical Science, Northeast Normal University, Changchun 130024, China; wangrr381@nenu.edu.cn
*    Correspondence: zhangy221@nenu.edu.cn; Tel.: +86-4318-5099-550

**Abstract:** In September 2020, the Chinese government proposed a climate change commitment that aims to make carbon emissions peak before 2030 and achieve carbon neutrality by 2060. In this context, it is important to examine the relationship between economic growth and carbon emissions. The Environmental Kuznets Curve (EKC) and decoupling analysis are commonly used assessment methods for regional sustainable development. Each method has a particular emphasis: the former focuses on long-term trends and the latter on short-term change. Integrating the EKC hypothesis with decoupling analysis is helpful to diagnose the relationship between economic growth and the carbon emissions of the manufacturing industry from the perspective of long-term trends and short-term changes. The results showed that the EKC passed the inflection point for both China's entire manufacturing industry and manufacture of nonmetallic mineral product subsector (MNM), but not in the other four main subsectors from 1995 to 2017. Strong decoupling, weak decoupling, and expansive coupling were observed between $CO_2$ emissions and the value added in China's entire manufacturing industry, in which weak decoupling accounted for the largest proportion. The decoupling index showed a downward trend on the whole. The decoupling status of subsectors from 1995 to 2017 was mainly weak decoupling, but different subsectors also showed characteristics of differentiation. At present, integrating EKC with decoupling has only occurred across the entire manufacturing industry and MNM. This study will provide suggestions for carbon reductions in China and will enrich the assessment methods of sustainable development.

**Keywords:** sustainability assessment; manufacturing industry; EKC; decoupling; China

## 1. Introduction

How to balance economic development and carbon emissions is an important issue that human society is facing in the 21st century, and it is also an important research direction for environmental regulation and sustainable development. Rapid economic growth has led to the large-scale use of fossil fuels, and global climate change is increasingly becoming a major threat to human beings [1]. Extreme temperatures and precipitation occur more frequently, ecosystems and water resources are being destroyed, environmental pollution is becoming more and more serious, and more indirect losses are caused [2]. In this context, sustainable development has become an important issue of concern in the international community. The practice of sustainable development occurs through the management of many governments from concepts to operations, such as cleaner production, carbon emissions trading, ecological compensation, and the green supply chain [3]. Therefore, it is very important to find an effective method for assessing sustainable development in order to improve the mode of regional economic growth and promote the harmonious development of the human–land relationship.

China is the largest carbon emitter in the world, and is assuming its responsibility for reducing carbon emissions [4]. The Law of the People's Republic of China on Promoting

Clean Production was revised in 2012, an Environmental Protection Tax Law was issued in 2017, and the Ministry of Ecology and Environment was reorganized in 2018. In order to implement the Paris Agreement on Climate Change (2016), China has continuously improved its carbon trading regulations. In September 2020, China further proposed a climate change commitment that aims to make $CO_2$ emissions peak before 2030 and achieve carbon neutrality by 2060 at the 75th Session of the United Nations General Assembly. Manufacturing is the most prominent industrial sector producing carbon emissions in China [5]. As the leading industry of economic development, the extensive development of the manufacturing industry is accompanied by a large number of energy consumption and carbon emissions. During the time period of 1995–2017, $CO_2$ emissions in China's manufacturing industry increased by 216.98%, accounting for more than half of China's total carbon emissions [6]. Therefore, the study of the relationship between the development of China's manufacturing industry and $CO_2$ emissions has important theoretical and practical significance for achieving the goal of carbon neutralization.

Research on the assessment of sustainable development around economic growth and environmental pressure is a matter of international academic exploration. The Environmental Kuznets Curve (EKC) [7] and decoupling analysis [8] have been widely accepted and used to assess sustainable development practices [9]. The Environmental Kuznets Curve means that pollution increases at low income levels as per capita GDP increases, and decreases at high income levels as per capita GDP increases [7]. Many studies tested the EKC hypothesis using different environmental indicators from different countries or regions, and it was found that, in addition to the "inverted U" relationship between per capita income and environmental degradation, there was also an "N" or even an irregular shape [10]. The term "decoupling" was extended to the economic field by the Organization for Economic Cooperation and Development (OECD) in order to study the link between environmental hazards and economic products in 2002. At present, EKC and decoupling analysis are widely used to investigate whether the changes in the economic development and environmental impact of a country or region are synchronized, but these two methods are mostly studied separately in the literature. In recent years, some scholars have tried to link the two in order to carry out their research [11]. In addition, the UN Environment Programme (UNEP) pointed out in 2011 that there were three stages of economic growth and pressures on environment and resources. It was considered that, according to the different decoupling indexes (DI) and corresponding stages A, B, and C (in the EKC), shown in Figure 1, decoupling may be integrated with the corresponding stages of the EKC [12]. However, there are few empirical studies of sustainable development integrating EKC with decoupling [13].

The structure of this paper is organized as follows. Section 2 reviews the literature and proposes the potential contributions that this study may make. Section 3 describes the methodology and data. Section 4 presents the empirical results using EKC and decoupling to analyze China's manufacturing value added and carbon dioxide emissions. Section 5 contains further discussion. Section 6 concludes the study and provides the important policy implications.

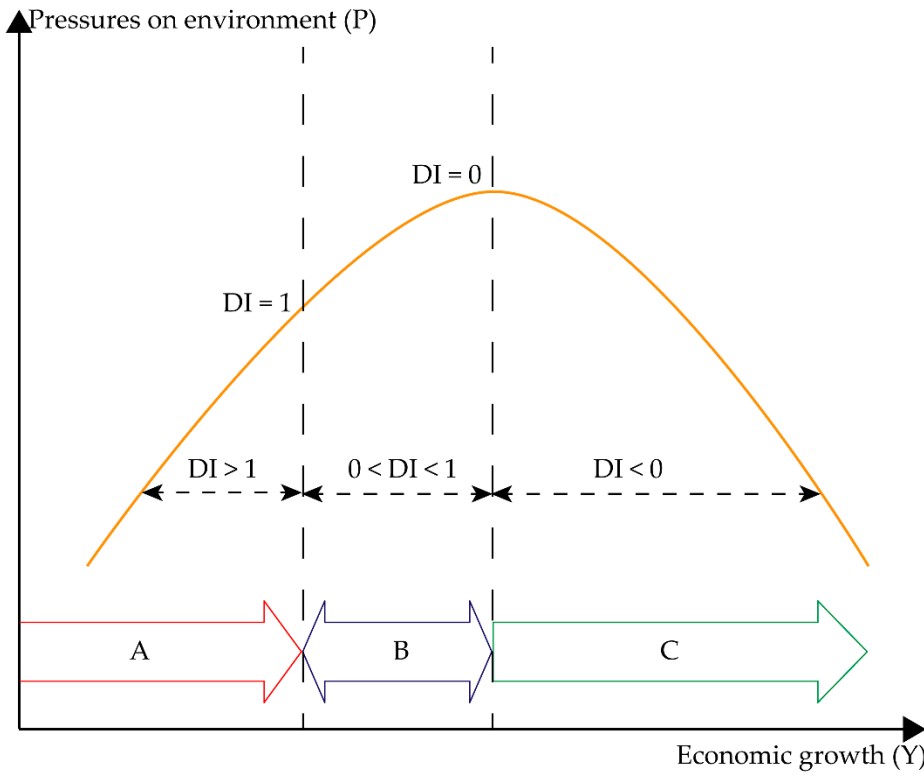

**Figure 1.** Three stages of economic growth and pressures on environment and resources. Source: [12].

## 2. Literature Review

### 2.1. EKC

In 1991, Grossman and Krueger proposed, for the first time, that the relationship between pollutants ($SO_2$ and soot) and per capita income was not linear, but "inverted U-shaped". In 1993, Panayotou named the "inverted U" between pollution and income the Environmental Kuznets Curve (EKC) [7]. The EKC hypothesis points out that, in the early stages of economic growth, environmental pollution will increase with the increase in per capita income, while it will be gradually reduced under the joint action of the structural and technical effects of economic activities and government environmental regulations in the medium and long term [14]. Many scholars have discussed the existence and shape of this hypothesis to varying degrees, but have not reached a consistent conclusion. The conclusions of the academic research on the EKC for carbon emissions can be roughly divided into three categories: (i) Ahmed and Long [15], Apergis and Ozturk [16], Alam et al. [17], Ozturk and Acaravci [18], and Ahmad et al. [19] and Churchill et al. [20] found the existence of an "inverted U" between carbon emissions and economic growth through empirical tests, and gave the inflection point; (ii) Holtz-Eakin and Selden [21], Friedl and Getzner [22], Kang et al. [23], Shahbaz et al. [24], and others believed that there was a long-term relationship between carbon emissions and economic growth, so this relationship was not an "inverted U" shape, but other shapes; (iii) Agras and Chapman [25], Azomahou et al. [26], Ajmi et al. [27], Al-Mulali et al. [28], Kaika and Zervas [29], and others disproved the existence of an EKC between carbon emissions and economic growth, and believed that there was, instead, a linear relationship between them. Other scholars believed that the shape of the EKC was related to the stage of economic development. For example, Han and Lu analyzed the relationship between per capita $CO_2$ emissions and per capita gross domestic product in 165 countries from 1980 to 2003 and found that there was an "inverted U" in countries with "high industry and high income", while there was only a weak "inverted U" in countries with "low industry and low income". The "low industrial and high income" countries had an "N", while the "high industrial and low income" countries showed an upward straight line with no inflection point [30].

Some scholars have begun to examine the EKC between $CO_2$ emissions and economic growth in specific sectors (such as industry, agriculture, transportation, services, etc.) [31–34], especially in the manufacturing industry. For example, Zhang et al. used the EKC model to calculate the turning points of carbon emission intensity, per capita carbon emissions, and carbon emissions of the manufacturing and construction industries in 121 countries, and found that a higher proportion of high-income countries reached inflection points [35]. Using the panel data of China's provinces from 2000 to 2013, Xu and Lin used a non-parametric additive regression model to explore the driving factors of $CO_2$ emissions in the manufacturing industry, and found that economic growth and $CO_2$ emissions in the manufacturing industry showed an "inverted U", mainly due to the fact that economic growth was driven by resources in the early stage and structural optimization in the later stage [36]. Again, Xu and Lin used a nonparametric additive regression model to test the main drivers of $CO_2$ emissions in China's iron and steel industry, and found that the nonlinear impact of economic growth on $CO_2$ emissions was consistent with the EKC [37]. Hidemichi et al. analyzed the relationship between $CO_2$ emissions and economic growth in different industries in OECD countries from 1970 to 2005, and found that the manufacturing industry did not observe an "inverted U" curve [38]. Nolen et al. found that there was a positive correlation between trade liberalization and pollution caused by the manufacturing industry by studying the relationship among pollution, income, and trade degree in 32 Mexican states, while the relationship between income and pollution was in agreement with the EKC [39].

### 2.2. Decoupling Analysis

"Decoupling" originally referred to the trends of changes between two physical quantities in the field of physics [40]. In 2002, the OECD first put forward the concept of "decoupling" and applied it to transportation, economic growth, energy consumption and manufacturing, as well as agricultural growth. It was proposed that decoupling can be divided into two states: absolute decoupling and relative decoupling, in which absolute decoupling showed that the economy grew but resource consumption remained constant or experienced negative growth, while relative decoupling showed that the economic growth rate was faster than the growth rate of resource consumption [8]. In 2005, Tapio studied the decoupling status between European transportation economic growth, transport volume, and $CO_2$ emissions from 1970 to 2001, and further put forward the concept of "decoupling elasticity" on the basis of the decoupling model proposed by the OECD [41]. Decoupling elasticity referred to the elasticity of carbon emissions, expressed as the ratio of economic growth to changes in carbon emissions, and appropriately reflected the sensitivity of changes in carbon emissions to economic growth. According to the value of decoupling elasticity, Tapio defined eight decoupling statuses, namely expansion negative decoupling, strong negative decoupling, weak negative decoupling, weak decoupling, strong decoupling, decline decoupling, growth connection, and decline connection. However, Tapio did not provide an accurate scientific basis for treating a change of DI = 1 ± 20% as coupling. In addition, a decoupling index of 0.8–1.2 actually included a wide-ranging combination of negative decoupling and weak decoupling, or a combination of recessive decoupling and weak negative decoupling. Compared with Tapio, Vehmas et al. abandoned the two combinations of coupling and put forward the theoretical framework of six possibilities of decoupling and linking with regard to environmental pressure and economic performance. Vehmas et al. divided the decoupling index into decoupling and coupling, then further subdivided it into six logical possibilities, namely weak decoupling, strong decoupling, recessive decoupling, weak coupling, strong coupling, and expansionary coupling [11]. The above three decoupling index types are mainstream tools to empirically analyze the decoupling effect of carbon emissions [42–45]. However, some scholars have proposed that there would also be differences in the level of economic development or pollution emissions in the same decoupling status. For example, cities with a high level of economic development could achieve relative decoupling or absolute decoupling by continuously

reducing pollution emissions, while cities with a low level of economic development could achieve the same decoupling status as cities with a higher level of economic development because of their low level of industrialization. Therefore, Xia and Zhong added per capita GDP to the decoupling criteria to comprehensively examine the decoupling status between urban economic development and environmental pollution [13].

In recent years, some scholars have begun to use different decoupling analysis methods to study the relationship between manufacturing development and $CO_2$ emissions. Hang et al. used the decoupling index proposed by Tapio to investigate the decoupling status between $CO_2$ emissions and value added in China's manufacturing industry and its subsectors from 1995 to 2015, and found that it was generally weakly decoupled, and the potential energy intensity was the main factor promoting the development of decoupling in the manufacturing industry [46]. Ren et al. found that China's manufacturing industry experienced stages of strong negative decoupling, weak decoupling, expansion negative decoupling, and weak decoupling, and concluded that the growth in economic output had the greatest impact on $CO_2$ emissions [47]. Dong et al. selected six major industrial sectors as research objects, introduced the Tapio decoupling model, and found that the proportion of value added of manufacturing was negatively correlated with carbon emissions, and that the manufacturing industry has experienced a process of decoupling–coupling–decoupling–coupling–decoupling [48].

Others have studied the relationship between development and $CO_2$ emissions in the manufacturing subsector. Zhou et al. used the decoupling analysis method proposed by the OECD to study the decoupling status and causes of carbon emissions and economic growth in China's power industry based on the logarithmic mean Divisia index (LMDI) model [49]. Wan et al. found that the decoupling relationship between carbon emissions and economic growth in China's equipment manufacturing industry was weak and showed a downward trend year on year from 2000 to 2014 [50]. Liu et al. analyzed and predicted the decoupling index between China's nonferrous metal consumption and GDP growth, and found that the total nonferrous metal consumption was not decoupled from GDP growth [51].

### 2.3. Integrating EKC with Decoupling

According to the stage of economic development, the UN Environment Programme (UNEP) put forward a three-stage theoretical diagram of economic growth and pressures on the environment and resources [12], arguing that, in the early stage of economic development, resource consumption and environmental degradation increased rapidly with economic growth, then DI $\geq$ 0, which was in the "climbing stage" of the EKC ((A) in Figure 1). In the middle stage of economic development, the growth rate of resource consumption and pollutant emissions was lower than that of economic growth, and DI ranged from 0 to 1 ((B) in Figure 1). In the late stage of economic development, while the economy continued to grow, resource consumption or pollutant emissions decreased, then DI < 0, which was in the "declining stage" of the EKC ((C) in Figure 1). After that, some scholars integrated EKC with decoupling for empirical analysis [52,53].

To sum up, EKC hypothesis and decoupling analysis are two powerful tools to study sustainability assessment. These two methods have been widely used and some progress has been made with them. However, there are still some limitations. The decoupling index can measure the degree of decoupling between $CO_2$ emissions and economic growth in the short term, but cannot determine the long-term trend. In contrast, the EKC hypothesis can explore the long-term relationship between $CO_2$ emissions and economic growth, but cannot capture their interaction in a particular year or in a particular stage. These two methods have been used in different studies. Even if the two methods are used simultaneously in the same study, their combination is barely discussed. In addition, most studies explore the decoupling and EKC between economic growth and carbon emissions at the macro level, such as in countries, regions, or specific provinces, but there are few studies that measure the existence of EKC and decoupling between carbon emissions and value added in the manufacturing industry and its subsectors in developing countries during the transition

period from the industry level [54]. The effective combination of EKC hypothesis and decoupling analysis is helpful to systematically investigate the relationship between the economic growth and carbon emissions of the manufacturing industry from the perspective of long-term trends and short-term changes, and judge the long-term sustainability and the stability of the decoupling of the manufacturing sector. This not only enriches the methods of sustainable assessment available, but also provides an improvement direction for the sustainable development of the manufacturing industry, accelerating the realization of the carbon neutrality goal.

Figure 2 is a flowchart of empirical analysis steps.

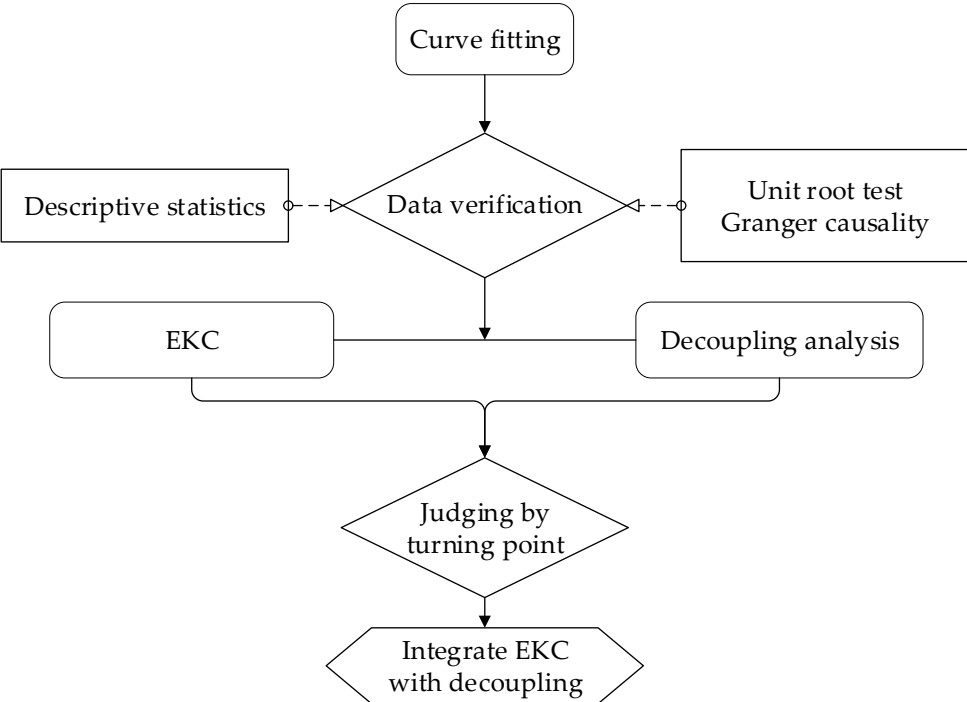

**Figure 2.** Steps of empirical analysis.

By introducing a three-stage theoretical diagram of economic growth and pressures on the environment and resources, put forward by UNEP [12], this study uses the method of decoupling analysis and EKC to explore the dynamic relationship between $CO_2$ emissions and the value added in China's entire manufacturing industry and its subsectors using the industry data of China's manufacturing industry and its subsectors. On the one hand, this can verify the existence of the EKC hypothesis and decoupling integration. On the other hand, it is expected that this study will provide a scientific reference to formulate targeted emission reduction policies.

## 3. Method and Data Source

### 3.1. EKC

In order to improve the accuracy of the EKC model estimation, this paper first sets the EKC model in the form of a cubic polynomial, based on the theoretical model of Selden and Song [55]. The cubic term will be eliminated if the coefficient of the cubic term is not significant and the cubic polynomial will be changed to a quadratic polynomial. Similarly, the quadratic term will be eliminated if the equation coefficient of the quadratic term is not significant and the quadratic polynomial is changed to a linear equation. According to the different equation forms, there can be different EKC shapes; for example, for a cubic polynomial, the shape should be "N" or "inverse N". For a quadratic polynomial,

the shape should be "inverted U" or "U". This paper refers to the analytical thinking of the Environmental Kuznets Curve proposed by [56]. The basic form of the model is as follows:

$$C_t = \beta_0 + \beta_1 G_t + \beta_2 G_t^2 + \beta_3 G_t^3 + \varepsilon_t \tag{1}$$

where $C_t$ is the $CO_2$ emissions in year $t$; $G_t$ represents the value added of the manufacturing industry in year $t$; $\beta_i$ is the coefficient of the explanatory variable to be estimated; and $\varepsilon_t$ is a random error term.

### 3.2. Decoupling

Based on the research of Vehmas et al. [11] and Tapio [41], this paper defined a decoupling index (see Table 1) to investigate the relationship between $CO_2$ emissions and value added in the manufacturing industry. It is calculated as follows:

$$DI_i = \frac{\Delta C}{\Delta G} = \frac{(C_t - C_0)/C_0}{(G_t - G_0)/G_0} \tag{2}$$

where $DI_i$ represents the decoupling index of $CO_2$ emissions and the value added in the manufacturing industry and its five high-emission subsectors in period $i$. $\Delta C$ represents the change rate of $CO_2$ emissions. $\Delta G$ represents the change rate of value added. $C_0$ and $C_t$ represent the $CO_2$ emissions in the initial and final years, respectively, and $G_0$ and $G_t$ represent the value added in the initial and final years, respectively.

### 3.3. Data Sources

The carbon emissions of China's manufacturing industry (in millions of tons) are estimated with reference to the carbon emissions calculation guidelines published by the IPCC and the relevant parameters published by the Chinese authorities. The formula is from [57]:

$$C = \sum_{i=1}^{i=18} E_i \times f_i (i = 1, 2, \ldots, 18) \tag{3}$$

where $C$ represents carbon emissions; $i$ represents the type of energy, including 18 types of energy such as raw coal, clean coal, crude oil, gasoline, fuel oil, liquefied petroleum gas, natural gas, electricity, and so on. $E_i$ is the consumption of $i$-type fuel after conversion into the standard coal equivalent and the fuel units are $10^4$ tons or 100 million cubic m. $f_i$ represents the emission factor of the $i$-type energy, which is calculated with reference to existing research.

The data on the energy consumption of different industries in China's manufacturing industry were from the China Energy Statistics Yearbook (1996–2018) and the data of the value added of different industries in China's manufacturing industry from 1995–2007 were from the China Statistical Yearbook (1996–2008). The National Bureau of Statistics of China has not released industrial value added since 2009, but it has published its annual, quarterly, and monthly growth rates, so the data for 2009–2017 were calculated on annual growth rates. In order to ensure the comparability of the data, the value added of the manufacturing industry was reduced to constant prices in 2000. According to the proportion of energy consumption and carbon emissions in five subsectors of the manufacturing industry, including smelting and pressing of ferrous metals (SFM), manufacture of raw chemical materials and chemical products (MRC), manufacture of nonmetallic mineral products (MNM), processing of petroleum, coking and processing of nuclear fuel (PPC), and smelting and pressing of nonferrous metals (SNFM), these five subsectors accounted for more than 70% for the period from 1995 to 2017, which were typical sectors with high energy and high emissions (Figure 3).

**Table 1.** Decoupling statuses and their classification.

| Decoupling State | Decoupling Type | Environmental Pressure | Economic Growth | DI | Significance |
|---|---|---|---|---|---|
| Negative decoupling | Expansive coupling | >0 | >0 | DI > 1 | Economic growth occurs at the cost of accelerated environmental destruction. |
| | Strong coupling | >0 | <0 | DI < 0 | The worst state when the economic is in a recession, while environmental pollution has increased. |
| | Weak coupling | <0 | <0 | 1 > DI > 0 | Low energy efficiency, which means the pollutant reduction rate is slower than the economic recession. |
| Decoupling | Weak decoupling | >0 | >0 | 1 > DI > 0 | Energy efficiency has improved, namely the energy consumption or pollutant emissions growth rate is slower than the economic growth rate. |
| | Strong decoupling | <0 | >0 | DI < 0 | The ideal state in which the economy grows, while environmental pressure decreases. |
| | Recessive decoupling | <0 | <0 | DI > 1 | Pollutant reduction rate is faster than economic recession. |
| Critical state | The critical point of coupling and weak decoupling | >0 | >0 | DI = 1 | The rate of economic growth is equal to the rate of the increase in environmental pressure. |
| | The critical point of weak decoupling and strong decoupling | =0 | >0 | DI = 0 | The rate of the increase in environmental pressure is equal to 0. |

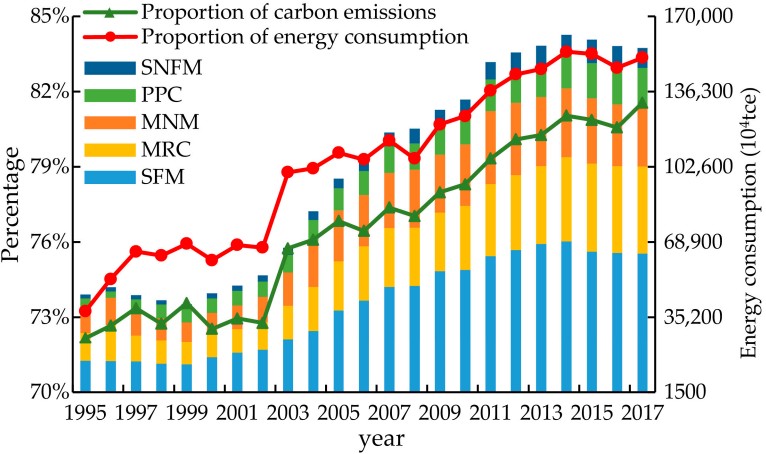

**Figure 3.** The proportion of carbon emissions and energy consumption of the five subsectors in the entire manufacturing industry.

In addition, the Circular of the General Office of the National Development and Reform Commission regarding the key work made in the initiation of the national carbon emissions trading market, issued in 2016, also proposed bringing eight key emission industries, such as iron and steel, petrochemical, chemical, nonferrous metals, etc., into the

first phase of the national carbon emissions trading market [58]. Therefore, this paper selects the five manufacturing subsectors mentioned above as its research objects.

## 4. Results

### 4.1. EKC Fitting

In order to test whether there is an "inverted U" between $CO_2$ emissions and value added in the manufacturing industry, we used historical experience data to observe their fitting curve, including the time series data of China's manufacturing industry from 1995 to 2017 and time series data of the five high-emission subsectors of the manufacturing industry from 1995 to 2017, in order to investigate the existence of Environmental Kuznets Curves in the manufacturing industry.

A fitting analysis of the relationship between the economic growth and $CO_2$ emissions of China's manufacturing industry showed that there was indeed an "inverted U" between economic growth and $CO_2$ emissions in the manufacturing industry. Before the inflection point of the "inverted U" is reached, $CO_2$ emissions would increase with the acceleration of the economic growth of the manufacturing industry and would show a downward trend with the economic growth of the manufacturing industry after reaching the turning point (Figure 4). An inflection point appeared when the value added of the entire manufacturing industry was about 20 trillion yuan, which preliminarily confirmed the existence of an EKC. By fitting the EKC of five high-emission subsectors of the entire manufacturing industry (Figure 5), it can be found that the $CO_2$ emissions and value added of nonmetallic mineral products (MNM) basically conformed to the "inverted U", in which an inflection point appeared when the value added of the entire manufacturing industry was about 800 billion yuan, while other subsectors have not shown an obvious "inverted U" and their $CO_2$ emissions all showed a trend of increasing with the increase in the value added. In addition, there were differences in carbon emission intensity in different subsectors, among which carbon emission intensity was the largest in SFM, but smallest in SNFM. It can be seen that there is still room for further improvement of SFM's technical transformation.

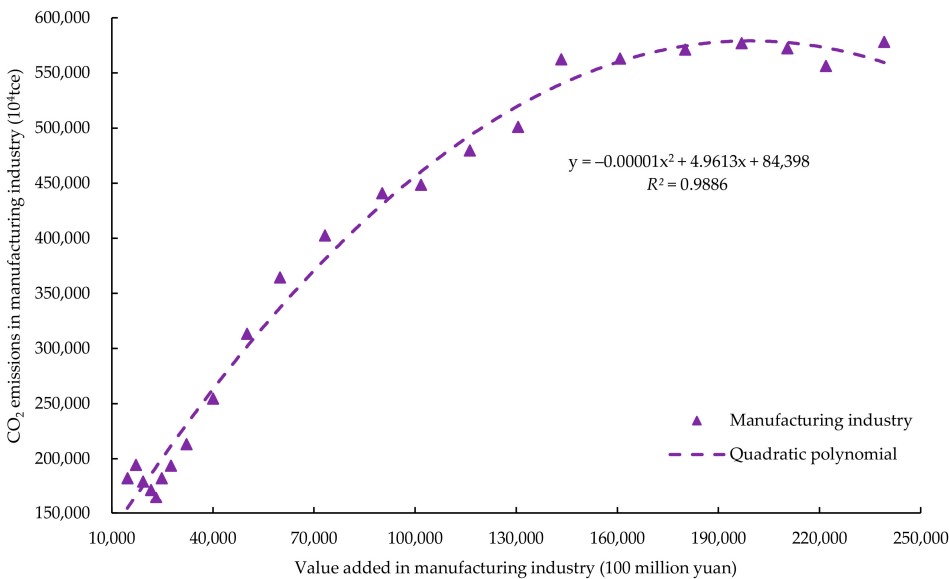

**Figure 4.** Fitting curve of $CO_2$ emissions and value added in China's manufacturing industry.

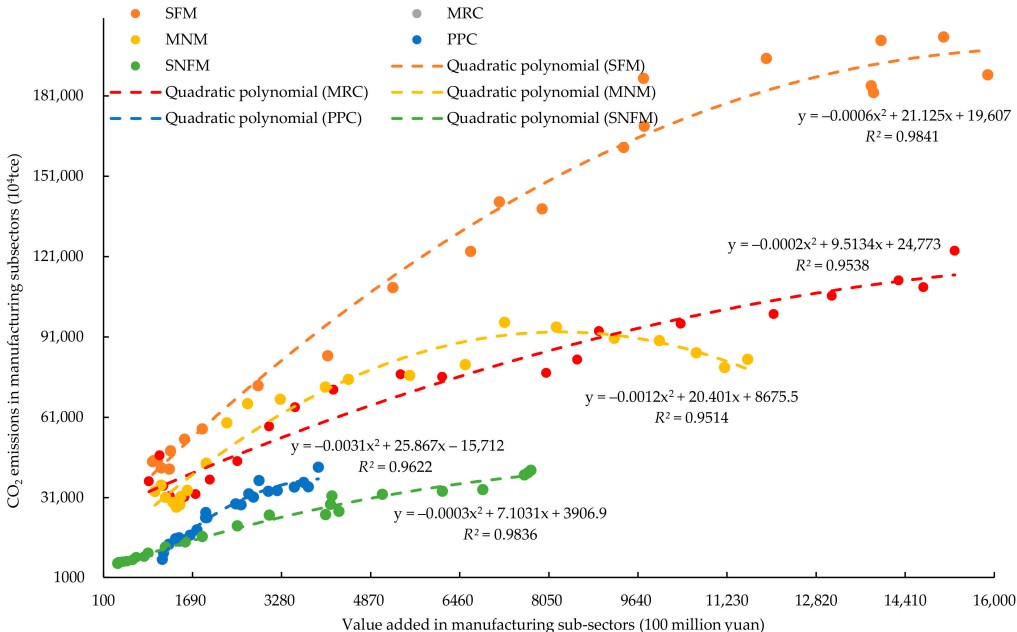

**Figure 5.** Fitting curve of $CO_2$ emissions and value added of five high-emission subsectors in China's manufacturing industry.

Since the fitting result is only a correlation, not necessarily a causal relationship, it cannot be concluded that the EKC existed in the manufacturing industry before the theoretical analysis, so it was only "possible". On the basis of the theoretical analysis, this paper used empirical analysis to further test the existence and rationality of the EKC between $CO_2$ emissions and value added in China's manufacturing industry.

### 4.2. Data Verification

#### 4.2.1. Descriptive Statistics

Table 2 reports the descriptive statistical results of the variables. Whether it is an explanatory variable or an explained variable, the mean is greater than the standard deviation, indicating that the degree of data dispersion is not high, and further analysis can be carried out.

**Table 2.** Descriptive statistics of variables.

| Variable | N | Mean | Sd | Min | Max | Skewness | Kurtosis |
|---|---|---|---|---|---|---|---|
| CM | 23 | 376,835.10 | 166,334.90 | 164,348.80 | 578,127.70 | −0.06 | 1.33 |
| CSFM | 23 | 118,196.70 | 63,245.51 | 41,642.36 | 202,933.50 | 0.01 | 1.33 |
| CMRC | 23 | 68,376.31 | 30,567.44 | 29,179.52 | 123,178.60 | 0.19 | 1.68 |
| CMNM | 23 | 61,274.24 | 25,157.87 | 27,436.33 | 96,382.08 | −0.16 | 1.41 |
| CPPC | 23 | 25,003.06 | 9933.46 | 7654.26 | 42,314.61 | −0.08 | 1.76 |
| CSNFM | 23 | 20,931.82 | 12,304.44 | 6142.55 | 41,207.83 | 0.24 | 1.60 |
| GM | 23 | 95,422.85 | 75,454.08 | 14,620.22 | 239,138.60 | 0.56 | 1.89 |
| GSFM | 23 | 6850.99 | 5324.77 | 978.27 | 15,884.10 | 0.32 | 1.63 |
| GMRC | 23 | 6147.07 | 5035.55 | 912.07 | 15,294.25 | 0.59 | 1.87 |
| GMNM | 23 | 4766.86 | 3741.94 | 1019.47 | 11,598.99 | 0.63 | 1.87 |
| GPPC | 23 | 2307.64 | 905.39 | 1152.91 | 3936.93 | 0.36 | 1.79 |
| GSNFM | 23 | 3164.06 | 2637.96 | 355.18 | 7729.27 | 0.54 | 1.88 |

Note: The first letter in each variable, namely "*C, G*" represents carbon emissions and value added, respectively, and the others are the abbreviations for manufacturing industry and its subsectors. For example, *CM* represents the carbon emissions of the manufacturing industry, and *GM* represents the value added of the manufacturing industry.

### 4.2.2. Statistical Test

### Unit Root Test

In order to prevent sequence spurious regression, the stationarity of the sequence must be tested before the regression. The ADF test is the most commonly used unit root test. When the error term has autocorrelation in the form of $p$-order autoregression, the ADF test can add the lag term in the regression formula to correct it. In addition, when the sample size is small, the ADF test is more robust [59]. In this paper, the ADF test method is used to test the stationarity of the data with the help of Eviews10, and the results are shown in Table 3. If the explanatory variables and the explained variables are nonstationary series, further differential processing is needed. Table 4 reports the results of the unit root test of variables after the first-order or second-order difference, which show that, after the second-order difference, $CO_2$ emissions disprove the original hypothesis of the existence of a unit root at a significance level of 1%, while the manufacturing value added disproves the original hypothesis of the existence of a unit root at a level of significance of 5%. Therefore, the variables selected in this paper are stationary series—that is, $CO_2$ emissions and value added in the manufacturing industry constitute the conditions for testing the long-term equilibrium relationship.

**Table 3.** Unit root test results of original variables.

| Variable | Inspection Type | ADF Test | Critical Values at Different Levels of Significance | | | $p$-Value | Conclusion |
|---|---|---|---|---|---|---|---|
| | (c,t,q) | Statistics | 1% | 5% | 10% | | |
| LnCM | (c,t,0) | −0.76 | −4.44 | −3.63 | −3.25 | 0.95 | Nonstationary |
| LnGM | (c,0,1) | −0.89 | −3.79 | −3.01 | −2.65 | 0.77 | Nonstationary |
| LnCMNM | (0,0,1) | 0.77 | −2.68 | −1.96 | −1.61 | 0.87 | Nonstationary |
| LnGMNM | (c,t,3) | −3.39 | −4.53 | −3.67 | −3.28 | 0.08 | Stationary |
| LnCMRC | (0,0,1) | 1.32 | −2.68 | −1.96 | −1.61 | 0.95 | Nonstationary |
| LnGMRC | (c,0,0) | −1.16 | −3.77 | −3.00 | −2.64 | 0.67 | Nonstationary |
| LnCPPC | (c,0,1) | −3.53 | −3.79 | −3.01 | −2.65 | 0.02 | Stationary |
| LnGPPC | (c,t,0) | −3.38 | −4.44 | −3.63 | −3.25 | 0.08 | Stationary |
| LnCSFM | (0,0,2) | 0.73 | −2.69 | −1.96 | −1.61 | 0.86 | Nonstationary |
| LnGSFM | (0,0,1) | 1.75 | −2.68 | −1.96 | −1.61 | 0.98 | Nonstationary |
| LnCSNFM | (0,0,0) | 5.63 | −2.67 | −1.96 | −1.61 | 1 | Nonstationary |
| LnGSNFM | (c,0,0) | −1.56 | −3.77 | −3.00 | −2.64 | 0.49 | Nonstationary |

Note: In the inspection type (c, t, q), c, t, and q represent the constant term, time trend, and lag order, respectively. The lag order is obtained according to the SIC criterion.

**Table 4.** Unit root test results of variables after difference.

| Variable | The Differential Order | Inspection Type | ADF Test | Critical Values at Different Levels of Significance | | | $p$-Value | Conclusion |
|---|---|---|---|---|---|---|---|---|
| | | (;c,t,q); | Statistics | 1% | 5% | 10% | | |
| LnCM | 1 | (c,0,0); | −2.18 | −3.79 | −3.01 | −2.65 | 0.22 | Nonstationary |
| LnGM | 1 | (c,t,0); | −1.55 | −4.47 | −3.64 | −3.26 | 0.78 | Nonstationary |
| LnCM | 2 | (c,t,0); | −6.31 | −4.50 | −3.66 | −3.27 | 0.00 | Stationary |
| LnGM | 2 | (c,t,0) | −3.83 | −4.50 | −3.66 | −3.27 | 0.04 | Stationary |
| LnCMNM | 1 | (0,0,0) | −2.26 | −2.68 | −1.96 | −1.61 | 0.03 | Stationary |
| LnCMRC | 1 | (c,t,4); | −3.90 | −4.65 | −3.71 | −3.30 | 0.04 | Stationary |
| LnGMRC | 1 | (c,0,0); | −3.12 | −3.79 | −3.01 | −2.65 | 0.04 | Stationary |
| LnCSFM | 1 | (c,t,3) | −2.86 | −4.57 | −3.69 | −3.29 | 0.20 | Nonstationary |
| LnCSFM | 2 | (0,0,0) | −8.09 | −2.69 | −1.96 | −1.61 | 0.00 | Stationary |
| LnGSFM | 1 | (c,0,0); | −3.01 | −3.79 | −3.01 | −2.65 | 0.05 | Stationary |
| LnCSNFM | 1 | (c,0,0); | −3.44 | −3.79 | −3.01 | −2.65 | 0.02 | Stationary |
| LnGSNFM | 1 | (c,0,0); | −2.63 | −3.79 | −3.01 | −2.65 | 0.10 | Nonstationary |
| LnGSNFM | 2 | (0,0,0) | −5.92 | −2.69 | −1.96 | −1.61 | 0.00 | Stationary |

Granger Causality Test

The Granger causality test is a commonly used method of econometric analysis, which is mainly used to test the "internal causality" and direction of time series, and is usually a pre-step in the testing of the EKC hypothesis [60]. After the unit root test is performed, it is necessary to carry out a Granger causality test on the variables in order to further analyze whether there is a correlation between the selected variables. The results in Table 5 show that the value added of the manufacturing industry is the Granger cause of $CO_2$ emissions at the 1% significance level, but that the opposite is not true. There is a causal relationship between the $CO_2$ emissions and value added of MNM and SNFM, but no causal relationship between the $CO_2$ emissions and value added of SFM.

**Table 5.** The results of the Granger causality test.

| Null Hypothesis | F Statistics | *p*-Value | Whether to Accept the Null Hypothesis |
|---|---|---|---|
| *LnG* is not the Granger reason for *LnC* | 6.62 | 0.01 | Reject |
| *LnC* is not the Granger reason for *LnG* | 2.59 | 0.11 | Accept |
| *LnGMNM* is not the Granger reason for *LnCMNM* | 4.09 | 0.04 | Reject |
| *LnCMNM* is not the Granger reason for *LnGMNM* | 3.68 | 0.05 | Reject |
| *LnGMRC* is not the Granger reason for *LnCMRC* | 10.87 | 0.00 | Reject |
| *LnCMRC* is not the Granger reason for *LnGMRC* | 0.24 | 0.79 | Accept |
| *LnGPPC* is not the Granger reason for *LnCPPC* | 3.36 | 0.08 | Reject |
| *LnCPPC* is not the Granger reason for *LnGPPC* | 0.60 | 0.45 | Accept |
| *LnGSFM* is not the Granger reason for *LnCSFM* | 1.11 | 0.36 | Accept |
| *LnCSFM* is not the Granger reason for *LnGSFM* | 1.991 | 0.17 | Accept |
| *LnGSNFM* is not the Granger reason for *LnCSNFM* | 3.00 | 0.08 | Reject |
| *LnCSNFM* is not the Granger reason for *LnGSNFM* | 3.59978 | 0.0512 | Reject |

*4.3. EKC Existence Test*

In this paper, the regression model is analyzed by using the national data from 1995 to 2017 with the help of SPSS20. When the multicollinearity in the model does not affect the significance of the variables concerned, the multicollinearity problem can be ignored [59]. Therefore, when estimating the quadratic or cubic equations of the value added in the entire manufacturing industry and its five high-emission subsectors, the cubic terms are removed if its coefficient is not significant. In Table 6, model 1 is the regression result of the cubic terms and model 2 is the result of re-estimation after removing the cubic terms. The regression result of $CO_2$ emission and value added in the manufacturing industry is shown in the following formula:

$$CM = 84,398.253 + 4.961GM - \left(1.244 \times 10^{-5}\right)GM^2 \tag{4}$$

**Table 6.** Estimation of the specific relationship between $CO_2$ emissions and value added of the entire manufacturing industry and its five high-emission subsectors.

| | Manufacturing Industry | | SFM | | MRC | |
|---|---|---|---|---|---|---|
| | Model (1) | Model (2) | Model (1) | Model (2) | Model (1) | Model (2) |
| Constant term | 76,651.03 *** | 84,398.25 *** | 31,101.49 *** | 19,605.02 *** | 15,777.31 *** | 24,774.09 *** |
| One-time term | 5.35 *** | 4.96 *** | 11.30 *** | 21.12 *** | 16.58 *** | 9.51 *** |
| Quadratic term | $-1.64 \times 10^{-5}$ * | $-1.24 \times 10^{-5}$ *** | 0.001 | $-0.001$ *** | $-0.001$ ** | $-0.0002$ *** |
| Cubic term | $1.09 \times 10^{-11}$ | | $-6.398 \times 10^{-8}$ ** | | $4.8 \times 10^{-8}$ ** | |
| Adjust $R^2$ | 0.987 | 0.987 | 0.987 | 0.982 | 0.957 | 0.949 |
| F statistics | 562.027 | 869.008 | 547.622 | 617.021 | 164.998 | 206.205 |
| | MNM | | PPC | | SNFM | |
| | Model (1) | Model (2) | Model (1) | Model (2) | Model (1) | Model (2) |
| Constant term | 3120.580 | 8681.291 ** | $-31,824.323$ ** | $-15,705.303$ *** | 2112.074 * | 3906.897 *** |
| One-time term | 25.357 *** | 20.401 *** | 48.393 *** | 25.864 *** | 10.263 *** | 7.103 *** |
| Quadratic term | $-0.002$ ** | $-0.0012$ *** | $-0.013$ * | $-0.003$ *** | $-0.001$ ** | $-0.0003$ *** |
| Cubic term | $5.426 \times 10^{-8}$ | | $1.27 \times 10^{-6}$ | | $8.49 \times 10^{-8}$ ** | |
| Adjusted $R^2$ | 0.947 | 0.947 | 0.96 | 0.958 | 0.985 | 0.982 |
| F statistics | 131.801 | 195.632 | 179.006 | 254.486 | 472.430 | 598.328 |

Note: *, **, and *** represent significance levels of 10%, 5%, and 1%, respectively.

The regression results show that the coefficient of the quadratic term is less than 0 and the first coefficient is greater than 0, so there is an "inverted U" between $CO_2$ emissions and value added in the manufacturing industry, indicating that the EKC in the manufacturing industry exists objectively. Combined with the above regression results, it is calculated that the reflection point of the "inverted U" appears in the position where the value added of the manufacturing industry is 19.939711 trillion yuan, which is consistent with the inflection point of the previous curve fitting to a certain extent. At present, the value added of China's manufacturing industry is 24.05054 trillion yuan, which has broken through the reflection point of the "inverted U". This indicates that China's $CO_2$ emissions are decreasing with the continuous increase in the value added. The model fitting results of the five high-emission subsectors of the manufacturing industry show that the subsectors all perform well in the quadratic regression model. In addition, the coefficients of the quadratic terms are all less than 0, and the openings are downward, which, to a certain extent, indicates that each subsector of the manufacturing industry is in accordance with the characteristics of the "inverted U". However, SFM, MRC, PPC, and SNFM are still on the left, rising stage of the "inverted U" at present, which highlights the pressures and challenges faced by the manufacturing subsectors with high emissions in the process of sustainable development.

### 4.4. Decoupling Index

Overall, there were only three types of decoupling between $CO_2$ emissions and value added in China's manufacturing industry from 1995 to 2017: strong decoupling, weak decoupling, and expansionary coupling (Table 7, Figure 6). Among them, weak decoupling accounted for the largest proportion, followed by strong decoupling in 1997–1999 and 2015–2016, while expansionary coupling only appeared in 2000 and 2011. Strong decoupling mostly occurred in 1996–1999, during which $CO_2$ emissions from China's manufacturing industry decreased by 301 million tons, or 15.47%, while the value added of the manufacturing industry increased by 35.12%. This shows that industrial development is significantly less dependent on energy consumption, mainly because China has implemented energy-saving and emission reduction policies. For example, the Law of the People's Republic of China on Conserving Energy, promulgated in 1997, and the Energy-saving management measures for key energy consuming units, promulgated in 1999, explicitly prohibited industrial projects that involved high energy consumption and the use of outdated technology and implemented the reform of state-owned enterprises, focusing on shutting down and transferring a large number of small emission-intensive enterprises. In addition, a large

number of township enterprises with development vitality were borne under the social and economic background of China's gradual transition in the 1980s. These township enterprises lacked an awareness of environmental protection and used outdated technology; thus, while they have made a great contribution to China's economic development, they have also become an important source of polluting emissions in China. Since 1996, the closure of a number of small enterprises with heavy pollution and poor efficiency has led directly to a significant reduction in carbon dioxide emissions [61].

**Table 7.** Decoupling status of China's manufacturing industry.

| Year | ΔC | ΔG | DI | Decoupling Status |
|------|------|------|------|-------------------|
| 1995–1996 | 0.07 | 0.17 | 0.38 | Weak decoupling |
| 1996–1997 | −0.08 | 0.12 | −0.64 | Strong decoupling |
| 1997–1998 | −0.04 | 0.12 | −0.35 | Strong decoupling |
| 1998–1999 | −0.04 | 0.07 | −0.58 | Strong decoupling |
| 1999–2000 | 0.11 | 0.07 | 1.54 | Expansive coupling |
| 2000–2001 | 0.06 | 0.11 | 0.57 | Weak decoupling |
| 2001–2002 | 0.10 | 0.17 | 0.60 | Weak decoupling |
| 2002–2003 | 0.19 | 0.24 | 0.79 | Weak decoupling |
| 2003–2004 | 0.23 | 0.25 | 0.92 | Weak decoupling |
| 2004–2005 | 0.16 | 0.20 | 0.83 | Weak decoupling |
| 2005–2006 | 0.10 | 0.22 | 0.47 | Weak decoupling |
| 2006–2007 | 0.10 | 0.23 | 0.41 | Weak decoupling |
| 2007–2008 | 0.02 | 0.13 | 0.13 | Weak decoupling |
| 2008–2009 | 0.07 | 0.14 | 0.49 | Weak decoupling |
| 2009–2010 | 0.04 | 0.12 | 0.36 | Weak decoupling |
| 2010–2011 | 0.12 | 0.10 | 1.25 | Expansive coupling |
| 2011–2012 | 0.00 | 0.12 | 0.01 | Weak decoupling |
| 2012–2013 | 0.01 | 0.12 | 0.12 | Weak decoupling |
| 2013–2014 | 0.01 | 0.09 | 0.11 | Weak decoupling |
| 2014–2015 | −0.01 | 0.07 | −0.12 | Strong decoupling |
| 2015–2016 | −0.03 | 0.05 | −0.51 | Strong decoupling |
| 2016–2017 | 0.04 | 0.08 | 0.50 | Weak decoupling |

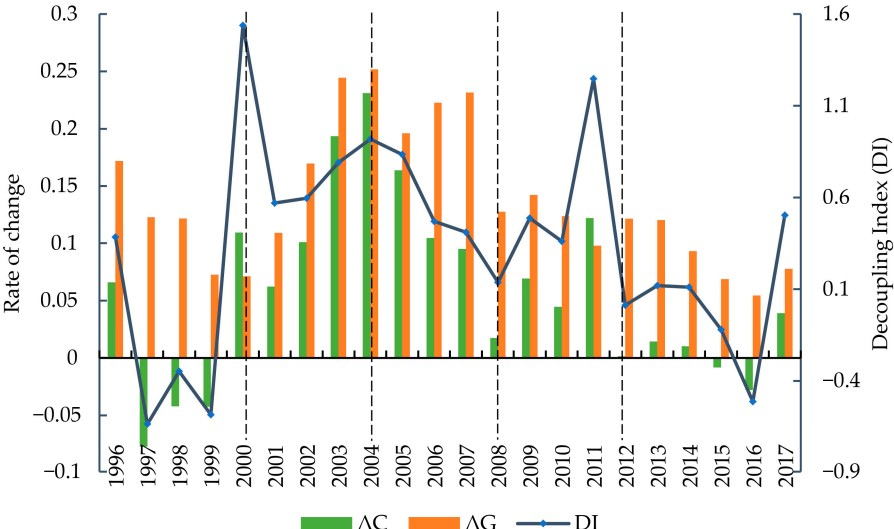

**Figure 6.** The change in decoupling index of China's manufacturing industry from 1996 to 2017.

Since joining the WTO in 2001, the trend of China's heavy industrialization has reappeared. In particular, since 2002, China's urbanization process has been comprehensively promoted, with a sharp increase in dependence on heavy industries such as cement, the chemical industry, and steel. The expansion of energy-intensive industries has stimulated

the increase in $CO_2$ emissions in the manufacturing industry. In 2004, the change rate of $CO_2$ emissions reached the maximum, and the change rate of value added in the manufacturing industry also reached its maximum, while the decoupling status was always weak decoupling. Since the implementation of the Eleventh Five-Year Plan, the central and local governments have paid more attention to energy conservation and emission reduction in the production process and put forward a series of binding targets for energy conservation and carbon emission reduction. Thus, the change rate of $CO_2$ emissions and the decoupling index continuously decreased from 2004 to 2008. From 2008 to 2011, the change rate of $CO_2$ emissions and the decoupling index increased continuously, and the decoupling status reached expansive coupling in 2011. Due to the government investing heavily in infrastructure during the economic crisis, high-carbon industries such as steel and cement developed rapidly in a short time. The decoupling status reached expansionary coupling in 2011. The delayed effect of the development of high-carbon industries on the environment is the main reason for the expansive coupling in 2011. After 2012, the decoupling type changed from weak decoupling to strong decoupling, showing an obvious low-carbon adjustment. The 18th National Congress of the Communist Party of China put forward the goal of "building a resource-saving and environment-friendly society" to stimulate local governments to increase restrictions on energy conservation and emission reductions in 2012. As a result, the growth rate of $CO_2$ emissions in the manufacturing industry slowed down after that, and $CO_2$ emissions even showed negative growth, resulting in the strong decoupling in 2015–2016.

After analyzing the decoupling status of the manufacturing industry, this paper further quantified the decoupling indexes of five high-emission subsectors from 1995 to 2017, and determined the decoupling status of each stage (Figure 7). The decoupling status of subsectors from 1995 to 2017 was mainly weak decoupling, but different sectors also showed characteristics of differentiation. Among them, SNFM was mainly weak decoupling, but showed expansive coupling in some years. PPC presented characteristics of large, repeated fluctuations, mainly by expansive coupling, strong coupling, weak decoupling, and strong decoupling. The decoupling status of SFM, MRC, and MNM from 1995 to 1999 was mainly strong decoupling, which showed a consistent trend with the manufacturing industry, while PPC showed a certain coupling situation at this stage. MNM showed an obvious trend of strong decoupling from 2012 to 2016, while other subsectors were dominated by weak decoupling during this period. In summary, among the five high-emission subsectors of the manufacturing industry, there was no stable decoupling status between $CO_2$ emissions and value added in the manufacturing industry, indicating that the decoupling status of China's manufacturing industry was still affected by many uncertainties.

*4.5. Analysis of Integrating EKC with Decoupling*

From the perspective of sustainable development, the decoupling index showed a downward trend with economic growth. This paper only analyzed the decoupling and EKC integration of the entire manufacturing industry and MNM, which were in accordance with the "inverted U" and passed the inflection point. Firstly, the curve equation of the decoupling index and value added of the entire manufacturing industry and MNM was fitted; the results showed that the fitting effect of the curve equation was poor, and the variables failed to pass the significance test. Combined with the scatter plot, we could see that there were some outliers. Thus, the curve was refitted after removing the outliers from 1996–1999 and 2011. The adjusted $R^2$ of the manufacturing industry reached 0.54, and F was equal to 19.768. The adjusted $R^2$ of MNM reached 0.411, and F was equal to 11.478 (Table 8). As a result, the fitting effect of the model is acceptable and we can use it for further analysis.

When DI = 0, the value added of the manufacturing industry was 20.44164 trillion yuan, which was basically consistent with the value added of 19.939711 trillion yuan at the inflection point of EKC. To some extent, it was verified that the model of the decoupling index and the manufacturing value added was better. When DI = 1, the value added of

the manufacturing industry is negative, which is not consistent with the actual situation. Since the late 1990s, China's manufacturing industry has made profound progress in terms of energy efficiency and pollution emissions with vigorous economic development, so the change rate of $CO_2$ emissions in China's manufacturing industry is always lower than that of economic growth (except in 2000 and 2011). It is difficult to find the phenomenon of expansive coupling in consecutive periods (compared with Figure 1, which lacks stage A), which is consistent with the conclusion drawn by Hang et al. [46].

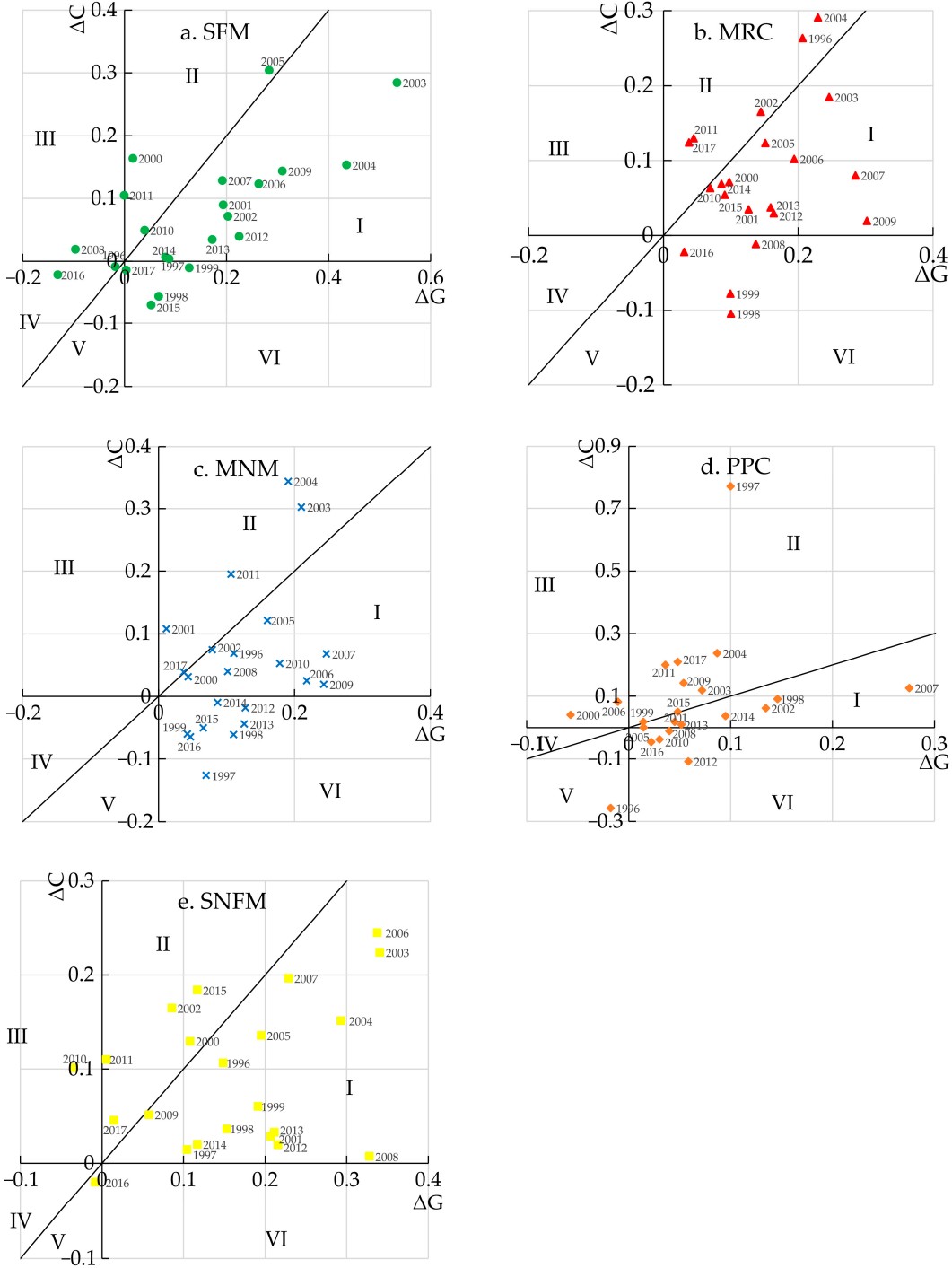

**Figure 7.** Decoupling distribution of five high-emission subsectors in the manufacturing industry. Note: I, II, III, IV, V, and VI represent weak decoupling, expansive coupling, strong coupling, weak coupling, recessive decoupling, and strong decoupling, respectively.

**Table 8.** OLS model of decoupling index and value added of the entire manufacturing industry and MNM.

|  | Manufacturing Industry | MNM |
|---|---|---|
| Constant term | 0.972 *** | 1.179 *** |
| One-time term | $-4.755 \times 10^{-6}$ *** | $-0.0001$ *** |
| Curve equation | $DI = 0.972 - 4.755 \times 10^{-6}G$ | $DI = 1.179 - 1.45 \times 10^{-4}G$ |
| Adjusted $R^2$ | 0.54 | 0.411 |
| F statistics | 19.768 | 11.478 |

Note: *, **, and *** represent significance levels of 10%, 5%, and 1%, respectively.

When DI = 0, the value added of the manufacturing industry is 842.143 billion yuan, which is basically consistent with the value added of 850.042 billion yuan at the inflection point of EKC. To some extent, it was verified that the model of decoupling index and manufacturing value added was better. When DI = 1, the value added of MNM is 127.857 billion yuan. Figure 8 shows the integration pattern diagram for decoupling and EKC of $CO_2$ emissions and value added in the entire manufacturing industry and MNM.

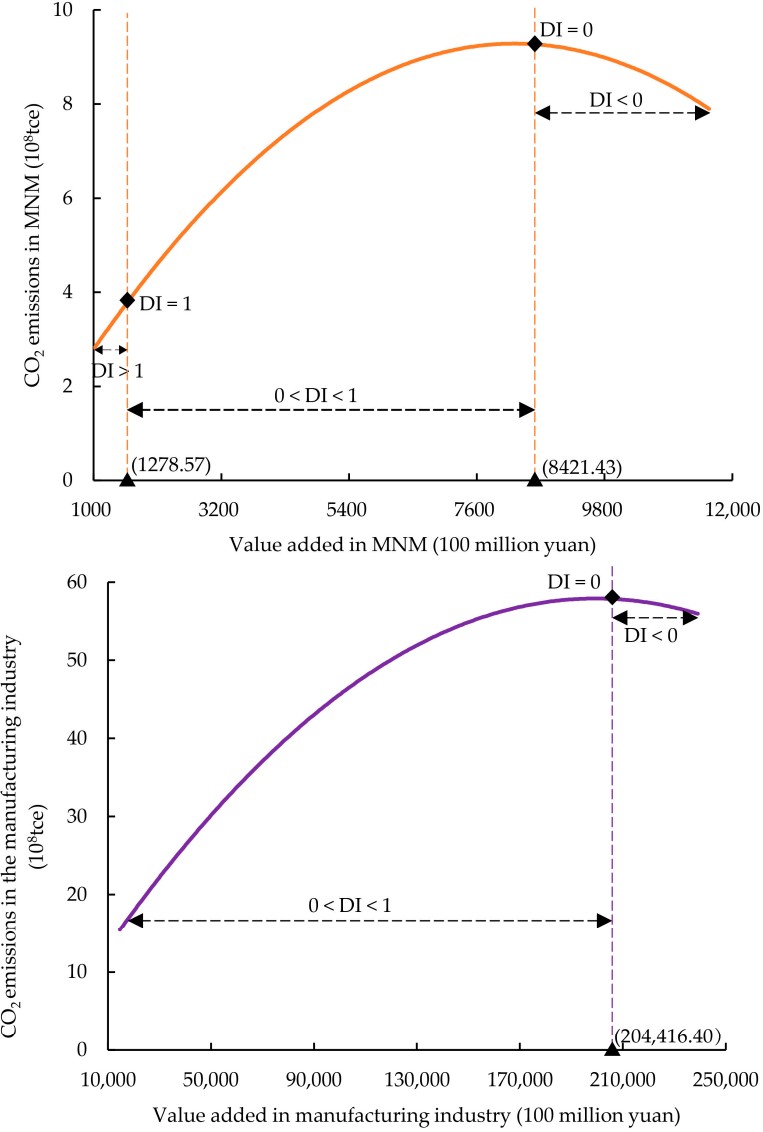

**Figure 8.** Decoupling and Environmental Kuznets Curve (EKC) of $CO_2$ emissions and value added in the entire manufacturing industry and subsectors: a coupling model diagram.

## 5. Discussion

(1) The EKC theory holds that there is an "inverted U" between economic development and environmental pollution. Baldwin proposed that this relationship is mainly attributed to the transformation of the industrial structure—that is, with the economic development, the leading industry has gradually shifted from the agricultural sector, with low emissions, to the industrial sector, with high emissions, and then to the service sector, with low emissions [62]. Therefore, the EKC theory is not necessarily suitable for specific industries (such as manufacturing). Our study found that the EKC hypothesis was significantly accurate in the manufacturing industry, which was inconsistent with the conclusions drawn by Zhang and other scholars [54]. We believed that, at the beginning of economic development, the value added of the manufacturing industry is significantly negatively correlated with environmental quality. However, with economic growth, people have a greater motivation and ability to reduce pollution emissions through the improvement of emission reduction technology, which leads to the inflection point of EKC in the manufacturing industry [63]. At the same time, we found that there was still no EKC law in some manufacturing subsectors, such as SFM, MRC, PPC, SNFM, etc., so these sectors still showed a monotonous increase in pollution emissions with the increase in value added. Previous studies have confirmed that the direct carbon emissions of these five high-emission sectors were decreasing, while the indirect carbon emissions were increasing due to the greater demand from other industries for these subsector products [64,65]. In addition, MRC is not only a high-energy-consumption industry, but also has a great pulling effect on carbon emissions in other industries. Therefore, in addition to controlling and strengthening the energy savings and consumption reductions in these subsectors with high energy consumption, high emissions, and high pollution, we should also supervise and control the carbon emissions of high-tech industries with low direct consumption of fossil energy [66].

(2) Based on the OLS fitting model of the decoupling index and manufacturing value added, there was no inflection point at DI = 1 and DI > 1 in the manufacturing industry. Looking at the decoupling index of the manufacturing industry, we can see that the decoupling index of China's manufacturing industry only passed the inflection point of DI = 1 in 2000 and 2010, and there was no consecutive DI > 1 from 1995 to 2017. This means that China's manufacturing industry did not experience a large number of cases where the change rate of $CO_2$ emissions was much greater than that of value added, at least in 1995–2017. Based on the reverse trend of data in this period, there would not be such a stage. In addition, based on the decoupling index, we further reveal that the dynamic relationship between the carbon emissions and the value added of China's manufacturing industry is closely linked with China's economic development stage. This is mainly reflected in the fact that the decoupling status of manufacturing carbon emissions and value added always corresponds to a specific node of social and economic development, such as the fact that China joined the WTO in 2001, the financial crisis occurred in 2008, and China's economy entered a new normal in 2012 [46].

(3) This study found that the decoupling status of manufacturing subsectors was mainly weak decoupling, but different subsectors also showed characteristics of differentiation. Specifically, the decoupling degree of MNM is the largest, and it has roughly experienced the evolutionary process of "strong decoupling–weak decoupling–strong decoupling", which was basically the same as the manufacturing industry. MNM is a typical energy-consuming industry. Since the implementation of macro-control over the cement industry in 2003, the state has continued to strengthen the control of high energy-consuming industries and issued a series of policy documents to strictly control high energy-consuming projects such as cement and refractories and to eliminate regressive production capacity [67]. On the whole, the decoupling status of SFM shows a cyclic evolution process of "strong decoupling–weak decoupling–expansibility coupling" On the one hand, this shows that the current development of SFM is still dominated by fossil energy consumption; it has not yet escaped the old correlation of "high energy consumption and high emission" [57]. On the other hand, the current relevant carbon emission

reduction policies and technical systems are not perfect, and the decoupling index has changed repeatedly. MRC is mainly weak decoupling, showing strong decoupling in some years, indicating that, to a certain extent, the current emission reduction policies could not completely offset the increase in carbon emissions caused by the increase in the output value in the process of economic growth. PPC and SNFM mainly showed expansionary coupling and weak decoupling during the study period, indicating that the change rate of carbon emissions was always higher than that of the value added. To a certain extent, this showed that the decoupling degree of PPC and SNFM was small, and should receive more attention in the future.

(4) The integration of EKC and decoupling analysis is helpful to further judge the trend through which the pollution index reaches the inflection point. Efforts to control carbon emissions need to be continuously strengthened in some sectors where coupling and decoupling occur alternately and where decoupling is unstable, such as improving the energy structure, enhancing energy utilization technologies, or carrying out carbon emissions trading [68]. It was found that the results of EKC in the entire manufacturing industry and MNM are consistent with the results of the decoupling index, which can be studied by using the three-stage theoretical diagram of economic growth and pressures on the environment and resources put forward by the United Nations Environment Programme (UNEP). However, the results of the three-stage theoretical diagram failed to materialize in the remaining four high-emission sectors. On the one hand, this was because the inflection point of the "inverted U" has not yet arrived in the other four manufacturing high emission subsectors; on the other hand, it was because the decoupling stability of these four high-emission manufacturing subsectors was poor. As a result, the long-term relationship between the decoupling index and economic value added could not be fitted. It can be seen that, if the energy structure of these four subsectors cannot be fundamentally changed or the energy use technology cannot be upgraded, the decoupling effect will continue to fluctuate significantly.

This paper only selects time series data at the national level in order to carry out a sustainable assessment based on EKC and decoupling analysis, which expands the research methods for sustainable assessment to a certain extent, but there is still room for further improvement. On the one hand, due to the incompleteness of the development stage and the instability of the decoupling index, the law of the trend has not yet appeared. Therefore, the integration of the EKC and decoupling analysis requires a long time span for some industries. On the other hand, the research scale of this paper is large, and its applicability to smaller areas is not clear. Different regions have different stages of development, and there are differences in the relationship between the EKC and the decoupling in manufacturing value added and $CO_2$ emissions. In the future, further studies on a smaller scale should be carried out to reveal the applicability of sustainable assessment methods based on EKC and decoupling analysis in different regions and in more detail.

## 6. Conclusions

The manufacturing industry is the pillar of China's industry and is a major carbon emitter. The focus of carbon emission reduction is the heavy industrial sector, with its high emissions, whose carbon emission reduction directly determines the realization of China's overall carbon emission reduction target. Based on the time series data of China's entire manufacturing industry and its five high-emission sectors from 1995 to 2017, this paper examined whether there was an EKC between $CO_2$ emissions and value added in the manufacturing industry and used the decoupling theory to analyze the short-term decoupling status between $CO_2$ emissions and value added. Finally, this paper comprehensively examined the internal relationship between the EKC hypothesis and decoupling theory through the derivation of a mathematical model. The main empirical results are as follows.

From 1995 to 2017, China's manufacturing industry and MNM passed the inflection point of the Environmental Kuznets Curve, while the existence of an EKC has not been

proven in the other four subsectors, which are still in the rising stage on the left side of the curve. This highlights the pressures and challenges faced by the high-emission manufacturing sectors in the process of sustainable development. At the same time, it also indicates that the carbon emission reductions in the manufacturing sector could not wait for the emergence of the inflection point and needed to make use of transformation and upgrading or the improvement of emission reduction technology in the manufacturing industry. There were only strong decoupling, weak decoupling, and expansionary coupling types of decoupling between $CO_2$ emissions and value added in China's manufacturing industry, in which weak decoupling accounted for the largest proportion, followed by strong decoupling, while expansionary coupling only appeared in 2000 and 2011. The decoupling index showed a downward trend on the whole, which indicated that the development of China's manufacturing industry was no longer occurring at the expense of an increase in $CO_2$ emissions. While weak decoupling was dominant in the subsectors, the stability of decoupling was poor. Different subsectors also showed characteristics of differentiation, indicating that the decoupling of China's manufacturing industry is still affected by many uncertainties. At present, integrating EKC with decoupling has only occurred in the entire manufacturing industry and MNM via mathematical analysis. Due to the poor decoupling stability, the integration analysis goal of the EKC and a decoupling method could not be achieved.

**Author Contributions:** D.L. designed and conceived this research; R.W. processed the data and wrote the paper; Y.Z. revised the paper. All authors have read and agreed to the published version of the manuscript.

**Funding:** This research was funded by the National Natural Science Funds of China, grant number Grant No. 41571115, 41771126 and 41571405.

**Data Availability Statement:** Not applicable.

**Conflicts of Interest:** The authors declare no conflict of interest.

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
