# Peer review of "Sustainability Assessment Based on Integrating EKC with Decoupling: Empirical Evidence from China"

_sustainability, doi:10.3390/su13020655_

Round 1

Reviewer 1 Report

This is an interesting article relevant for the readership of Sustainability journal. The authors are, however, advised to do some minor improvements in order to make it suitable for publishing. My advices are as follows:

  • do not finish chapters with figures (e.g., Fig. 2, Fig. 3 and others). In most instances this can be resolved by rearranging the text.
  • please use journal's template to format the manuscript, including list of references
  •  

Reviewer 2 Report

I feel that the concept of establishing the EKC and Coupling in tandem as possibly a universal measure of the success of an economic unit in achieving the absolute reduction of CO2 emissions (Paris Accord) is irresitable. Necessary. The paper mostly achieves the explanation of theory and demonstration of practice of this new combined measure.  Areas for improvement follow.

Abstract will require adjustment to reflect the changes set out in the following items. However it is written in a strong 'voice'.  Well done.

Introduction section does not explain the possible benefits of integrating EKC and coupling as hinted at in the Abstract.  I suggest that you add content based on that developed in lns 206-212.

Ln 58 onwards briefly explain the EKC tenet re environmental degradation/per capita income here

Method and data source.

Well done on Figure 5.  Clear explanation of what you found. Excellent communication.

Discussion section does not adequately address the efficacy of the combined EKC/Coupling measure to inform decisions aimed at achgieving absolte reduction in CO2 decisions.  More discussion of scombined strengths/weaknesses, how each complements the other.

Lns 536-542 This is a run-on sentence and as such it is difficult to understand what the authors intend to convey.  Rewrite as several smaller sentences for clarity.

Conclusion section has two large paragraphs (Lns 594 - 645) discussing possible actions to be taken but these were never the subject of the research. Indeed they have no underpinning sources/citations. Whilst these were foreshadowed in the Abstract they have not been developed through research and are properly the subject of subsequent important research. Delete.

Minor opportunities for improvement:

Ln 38 'and so on' is meaningless and unprofessional. Delete.

Ln 101 replace 'found' with 'who found'.

Lns 20-282 the words 'under 2017' are not suitably placed to achieve clear meaning. Either delete or add at the end of the sentence '...for the period 1995 to 2017.'

Ln 296 replace 'verify' with 'investigate'.  Verify would be used where somebody has already asserted that there is an inverted U curve.  I gather that you are the first to explore, to investigate this.

Reviewer 3 Report

The article addresses an important and current topic of sustainable development assessment at the macro level (China). Undoubtedly, this is an important step towards studying the relationship between CO2 emissions and value added in the manufacturing industry.

However, I have many comments to the manuscript of an editorial and substantive nature:

In the abstract, I would emphasize more the justification for taking the topic (background) and limit the description of the results obtained.

The introduction contains the following sentences:

Line 42-43: "China is the largest carbon emitter in the world, and is taking responsibility for reducing carbon emissions."

Line 50-51: "Manufacturing is the most prominent industrial sector producing carbon emissions in China."

Line 53-55: "During the time period of 1995-2017, CO2 emissions in China's manufacturing industry increased by 216.98%, accounting for more than half of China's total carbon emissions."

Please provide the sources of this information.

Line 82: "The reminder of this paper is organized as follows." - better to use the word "structure"

Line 86: "The final section concludes the study and provides the important policy implications." - it's better to write section 6 with reference to it

In the "Literature Review" section, please check the references very carefully - there are many errors:

Before (ii) and (iii), please use a semicolon, not a dot

Line 123: it's "Xu et al.", it should be Xu and Lin

Line 128: „they” – avoid personal forms, better "the same authors" or "Xu et al."

Line 130-131: "Fujii and Managi" appears in the text, but missing in references

Line 159: "Vehas et al." - missing in references

Line 162: "He" - avoid personal forms, better "the same authors" or "Vehas et al."

Line 172: it's "Xia et al.", it should be Xia and Zhong

Line 177: Ye et al. - appears in the text and the reference is [37]: Hang et al.

Line 188: Lu et al. - appears in the text and the reference is [40]: Wan et al.

[13] Alam et al.

[18] Kang et al.

[20] Azomahou et al.

[30], [31] – Xu and Lin

Section "Method and Data Source":

Line 233: Selden and Song

Line 242: the fomula number (1) should be on the same line as the formula

Line 251: the fomula number (2) should be on the same line as the formula

Table 1 - please improve the formatting

3.3. Data sources:

Please justify the choice of 5 subsectors more precisely and describe in more detail the basis on which the data used for statistical analyzes were approximated.

"The data come from the China Energy Statistical Yearbook (1996-2018)." - 2 times the same information on lines 270 and 274

"ADF test method is used" - please provide an explanation of why this test was selected, there are many others; was the selection of the test taken into account, for example, the distribution of data?

Granger causality test - justify the choice of this test

It would be good to write what software was used for statistical analysis: Statistica? Minitab? other?

Line 369: the fomula number (4) should be on the same line as the formula

Line 489: Ye et al. [51] - wrong reference, [51] is Hang et al.

Line 502: Discussion without a dot

Line 620-621: "build a beautiful China" - avoid such wording in the scientific text

Please avoid the personal form "we" etc.

Literature selected correctly, although there are some errors in references and some sources are repeated, e.g. [37] and [51]; requires careful verification and ordering; 2019-2020 sources can be added

The results and discussion were presented correctly, although it is not clear how the individual values were obtained. It would be good to describe how the individual results were generated. There are also no directions for possible further research and no information about the limitations of the statistical analysis. It was also possible to provide the application possibilities of the proposed approach (other countries? Sectors?).

The conclusions are general and should relate more to the results achieved and the latest literature.

The text certainly has some advantages, but I think that in order to publish it, it would be good to introduce all the suggested corrections.

I evaluate the text positively and it is written correctly from the scientific point of view. The comments are mostly polemical, but before publication, I think it would be good to make all the suggested corrections.
